# Structural basis for dolichylphosphate mannose biosynthesis

Rosaria Gandini[1], Tom Reichenbach [1], Tien-Chye Tan[1] & Christina Divne [1]

Protein glycosylation is a critical protein modification. In biogenic membranes of eukaryotes and archaea, these reactions require activated mannose in the form of the lipid conjugate dolichylphosphate mannose (Dol-*P*-Man). The membrane protein dolichylphosphate mannose synthase (DPMS) catalyzes the reaction whereby mannose is transferred from GDP-mannose to the dolichol carrier Dol-*P*, to yield Dol-*P*-Man. Failure to produce or utilize Dol-*P*-Man compromises organism viability, and in humans, several mutations in the human *dpm1* gene lead to congenital disorders of glycosylation (CDG). Here, we report three high-resolution crystal structures of archaeal DPMS from *Pyrococcus furiosus*, in complex with nucleotide, donor, and glycolipid product. The structures offer snapshots along the catalytic cycle, and reveal how lipid binding couples to movements of interface helices, metal binding, and acceptor loop dynamics to control critical events leading to Dol-*P*-Man synthesis. The structures also rationalize the loss of dolichylphosphate mannose synthase function in *dpm1*-associated CDG.

[1] School of Biotechnology, KTH Royal Institute of Technology, S-10691 Stockholm, Sweden. Correspondence and requests for materials should be addressed to C.D. (email: divne@kth.se)

Protein glycosylation is critical for protein folding, sorting, stability, and function in all domains of life[1–3]. In biogenic membranes (i.e., self-synthesizing endoplasmic reticulum (ER) membrane in eukaryotes and cytoplasmic cell membrane in archaea), the activated mannose used for protein glycosylation and glycosylphosphatidylinositol anchor synthesis[1–8] is provided in the form of the glycolipid intermediate dolichylphosphate β-D-mannose (Dol-P-Man). An integral membrane protein, Dol-P-Man synthase (DPMS; EC 2.4.1.83)[9], is responsible for Dol-P-Man biosynthesis by catalyzing the transfer of mannose from guanosine diphosphate α-D-mannose (GDP-Man) to dolichyl-phosphate (Dol-P). In eukaryotes, Dol-P-Man synthesis takes place on the cytosolic face of the ER membrane[10], followed by translocation of the glycolipid across the membrane to expose the activated mannosyl group to the ER lumen[11] where initial protein glycosylation and glycosylphosphatidylinositol biosynthesis take place. In archaea, Dol-P-Man is synthesized on the cytoplasmic face of the plasma membrane, and flipped to the cell exterior, i.e., the topological counterpart of the ER lumen, to serve as mannosyl donor for glycosylation of surface proteins of the paracrystalline glycoprotein cell envelope[12, 13]. Altered DPMS activity in humans is linked to cancer and congenital disorders of glycosylation[14–16].

Here we report the structure of a bona fide DPMS. High-resolution crystal structures of wild-type *Pyrococcus furiosus*

**Fig. 1** Overall structure of *Pf*DPMS. **a** Structure of *Pf*DPMS with bound GDP-Man and Mn$^{2+}$. The acceptor loop (*purple*) is locked in a closed conformation. **b** Two-fold pseudo symmetry of the TMD1/TMD2 interface. **c** Sequence similarity between TMD1 and TMD2, including the residues that define the pseudo twofold symmetry in **b**

DPMS (*Pf*DPMS) in complex with donor and acceptor substrates, as well as with Dol-*P*-Man product, were determined. The enzyme complexes provide snapshots along the catalytic cycle, and offer a detailed picture of how Dol-*P*-Man is synthesized in *Pyrococcus*. We expose a sophisticated interplay between juxta-membrane interface (IF) helices, donor, metal, and acceptor that controls Dol-*P*-Man synthesis. While the IF helices are strictly required for Dol-*P*-Man synthesis, the transmembrane (TM) domain is expendable. Furthermore, our structural data rationalize the importance of several amino acids that are implicated in human disease. Considering the importance of DPMS function for organism biology, including several common pathogens, these results provide a valuable structural platform for design of therapeutics that target Dol-*P*-Man-dependent protein glycosylation.

## Results

**Overall structure**. We determined the crystal structures of *Pf*DPMS in complex with metal-bound nucleotide (GDP•Mg$^{2+}$) and donor (GDP-Man•Mn$^{2+}$; Supplementary Table 1). The overall structure of *Pf*DPMS features an N-terminal, cytoplasmic catalytic domain covalently attached to a TM domain with four TM helices (TMHs) arranged as two TMH dimers, TMD1 and TMD2 (Fig. 1). The TM domains of DPMSs show varying topological complexity, and based on differences in the TM domain, two classes of DPMS have been proposed[17] (Supplementary Fig. 1a, b). The type-I enzymes are represented by DPMS from *Saccharomyces cerevisiae*[9], where the catalytic domain is covalently attached to a single C-terminal TMH (Supplementary Fig. 1a). In contrast, the human type-II DPMS forms a multi-subunit complex with a catalytic domain (DPM1) that lacks TMHs but associates with two separately encoded membrane proteins, DPM2 and DPM3, each of which comprises 80–90 amino acids that fold into two TMHs[18] (Supplementary Fig. 1b). The structure of the TM domain in *Pf*DPMS places the pyrococcal and thermococcal enzymes in a new class of DPMS, type-III (Supplementary Fig. 1c), which is topologically similar to the assembled mammalian DPM1/DPM2/DPM3 complex. Human DPM1 requires DPM3 for targeting to the ER membrane. In contrast, DPM2 enhances activity, although the exact role is unknown[18]. Topologically, DPM2 and DPM3 in human DPMS could correspond to the two TMH dimers in *Pf*DPMS.

The catalytic domain of *Pf*DPMS belongs to family 2 of inverting glycosyltransferases (GT2; www.cazy.org)[19] with a GT-A fold (Supplementary Fig. 2) where a conserved DXD sequence motif (Supplementary Figs. 2 and 3) coordinates the catalytic metal ion and donor substrate[20]. The canonical GT-A fold was first described for the spore-coat protein SpsA from *Bacillus subtilis*[21], which features a 7-stranded β-sheet flanked by α-helices on both sides (Supplementary Fig. 2). The GT-A fold can be further broken down into an N-terminal and C-terminal half. The N-terminal region is composed of a Rossmann-like four-stranded, parallel β-sheet (strands β1–β4) with two α-helices (α1 and α2) on one side of the sheet, and one α-helix (α3) on the other side. This Rossmann-like N-terminal motif extends into a C-terminal motif that folds as a three-stranded mixed β-sheet (strands β5–β7) with one α-helix on either side (α4 and α5). The β-strands of the N- and C-terminal halves form an extended β-sheet (Supplementary Fig. 2). In addition to the GT-A core structure, members of the GT2 family display various additional structural features relating to their unique functional requirements and subcellular localization. Besides the TM domain, two IF helices (IFH1 and IFH2) are appended to the cytoplasmic GT-A domain in *Pf*DPMS (Supplementary Fig. 2a). The IF helices are amphipathic, and oriented to expose their hydrophobic faces towards the membrane (Supplementary Fig. 4).

We note that the TM domain shows an unusual arrangement of the two TM dimers. TMD1 is perpendicular to the membrane and spans ~45 Å (Fig. 1a), while TMD2 is rotated by 60° with respect to TMD1, giving the impression of walking legs that associate at a junction. Several observations imply that this arrangement is authentic, rather than an artifact of detergent solubilization or crystal-packing interactions. First, the interface between TMD1 and TMD2 displays pseudo-two-fold symmetry at the junction (Fig. 1b), which is also manifested as sequence similarity between the TM dimers (Fig. 1c). The pseudo-twofold interface is optimally packed at the junction, and decreasing the relative angle between the TMDs would result in severe clashes, most seriously between Phe237 (TMH1) and Tyr342 (TMH4), Phe278 (TMH2) and Trp350 (TMH4), Phe278 (TMH2) and Phe301 (TMH3). Analysis of the interface between TMD1 and TMD2 using PISA (Proteins, Interfaces, Structures and Assemblies; http://www.ebi.ac.uk/pdbe/pisa/)[22] suggests a stable association in solution with a buried interface area of 1620 Å$^2$, and a complex formation significance score of 1.0. The estimated solvation free energy gain ($\Delta G^{int}$) is –12.6 kcal/mol for formation of the TMD1/TMD2 assembly as observed in the crystal structure, and the free energy of assembly dissociation ($\Delta G^{diss}$) is 8.6 kcal/mol, as predicted by PISA.

Second, we analyzed the packing interactions between crystallographically related *Pf*DPMS molecules to evaluate whether specific crystal contacts could be responsible for the observed TMD1/TMD2 arrangement. In the crystal, detergent-solubilized *Pf*DPMS molecules are arranged in an array forming a continuous layer of TMHs (Supplementary Fig. 5a, b) with detergent molecules (LDAO) interspacing the TM domains (Supplementary Fig. 5c, d). Two unique interfaces are formed in the crystal; TMH1/TMH1 (Supplementary Fig. 5c) and TMH3/TMH3 (Supplementary Fig. 5d), where the TMH3/TMH3 interface buries fewer LDAO molecules, and appears better packed. We analyzed the TMH1/TMH1 (TMD1) and TMH3/TMH3 (TMD2) interfaces using PISA, and as judged by the complex formation significance score values of 0.0, neither interface is expected to be stable. The results from the PISA analysis and the presence of LDAO molecules at the interfaces between crystallographically related molecules suggest that: (i) detergent-solubilized *Pf*DPMS crystallize as monomers and (ii) that the interactions between *Pf*DPMS molecules are weak and unlikely to be responsible for the tilted orientation of TMD2. Based on these observations, we propose that the observed arrangement of TMD1 and TMD2 is authentic, and that detergent-solubilized *Pf*DPMS exists mainly in monomeric form.

Finally, combined NMR and computational studies[23] suggest that the isoprenoid chain of Dol-*P* is oriented in the membrane with its long axis forming an angle to the bilayer interface of roughly 75°, which hints at a functional role of the tilt of TMD2 to provide a pre-organized binding surface for the dolichyl moiety of Dol-*P* and Dol-*P*-Man. Although this would probably be favorable energetically, especially at the extreme temperatures at which *P. furiosus* grows (70–103 °C)[24], it is apparently not a strict requirement for Dol-*P* binding since type-I DPMSs contain only a single TMH.

**Binding of donor substrate**. The guanosine moiety of GDP is located in a pocket formed at the C-terminal ends of the parallel β-strands β1–β4, where interactions are offered mainly with the β-α loop regions (Supplementary Fig. 2). Guanosine interacts with the protein through several possible hydrogen bonds: guanine O6 – Gly68 N (β3-α3 loop); guanine N1 and N2 – Asp39 OD2 (β2-α2 loop); ribose O2 – Tyr10 N (β1-α1 loop); and ribose O3 – Ala90 N (β4-α4 loop). In addition, the guanine ring stacks with the hydrophobic Leu69 side chain (β3-α3 loop). The

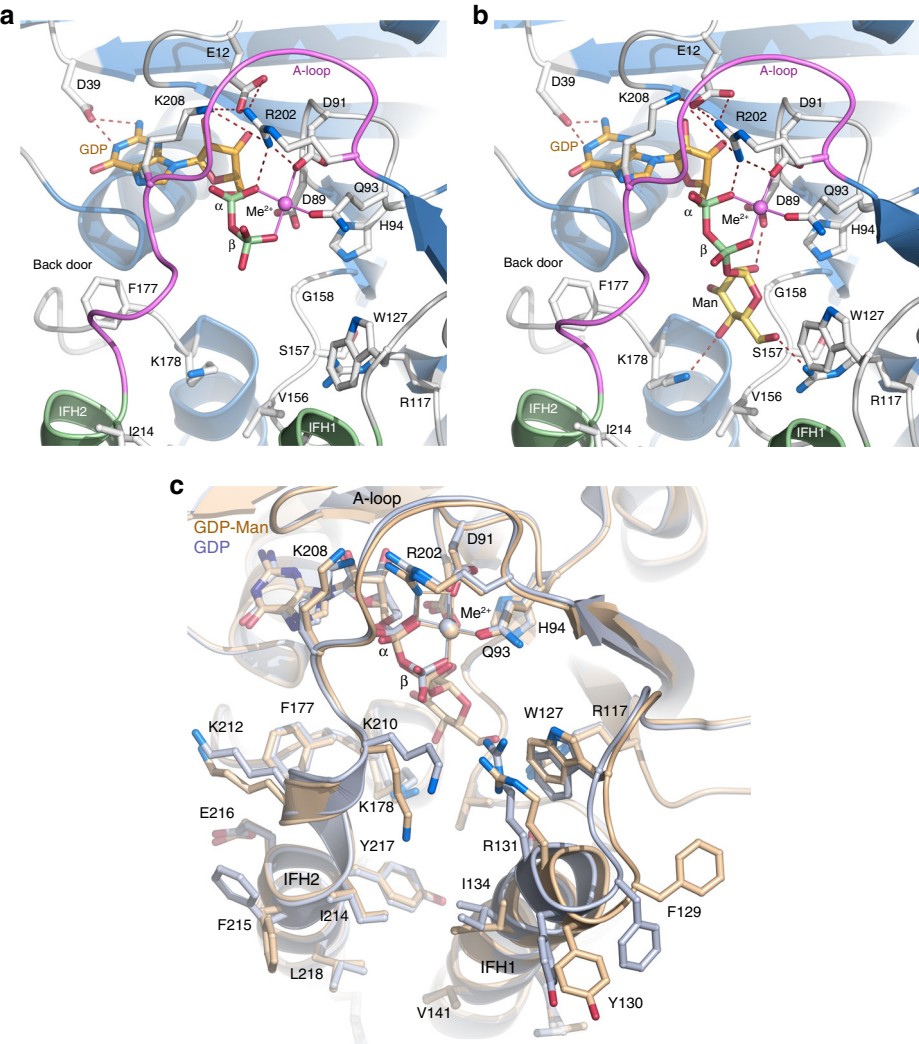

**Fig. 2** Details of nucleotide and donor binding. **a** The *Pf*DPMS active site with bound GDP•Mg$^{2+}$ and **b** GDP-Man•Mn$^{2+}$ in the absence of acceptor substrate. The acceptor loop (*purple*) is locked in a closed conformation by ionic bonds between Glu12 and Arg202 and Lys208 close to the GDP $\alpha$-phosphate. The IF helices do not participate directly in donor binding. Asp39 interacts with the N1 and N2 atoms of the guanine base, and Phe177 acts as a gate and closes off the pocket exit. **c** Superimposition of the *Pf*DPMS•GDP•Mg$^{2+}$ (*beige*) and *Pf*DPMS•GDP-Man•Mn$^{2+}$ (*light blue*) complexes

diphosphate-binding site is formed at the canonical DXD motif located in the β4-α4 loop, which is defined by DAD in DPMSs (Supplementary Fig. 3), and forms the inner wall of the diphosphate-binding site. The active site in the catalytic domain is gated by a flexible loop (residues 202–210), which we refer to as the acceptor loop, or A-loop (Figs. 1a and 2). In the *Pf*DPMS•GDP•Mg$^{2+}$ and *Pf*DPMS•GDP-Man•Mn$^{2+}$ complexes, the A-loop is closed and folded over the diphosphate-binding site to create an outer wall ("front door") of the active site (Fig. 2).

Asp91 and Gln93 in the $^{89}$DADLQH$^{94}$ motif coordinate the α- and β-phosphate groups of GDP *via* the metal cation: α-phosphate O2A – Me$^{2+}$ – Asp91 OD2, and α-phosphate O2B – Me$^{2+}$ – Gln93 OE1 (Fig. 2a, b). The conformation of the A-loop couples to donor binding through a network of interactions: GDP α-phosphate O2A – Arg202 Nh1, and α-phosphate O2B – Lys208 N. The A-loop is further secured by ionic interactions: Arg202 Nh2 – Glu12 Oe1; Arg202 Nh2 – Oe2; Arg202 Nh2 – Glu206 O; and Glu12 Oe2 – Lys208 Nz. On the other side of the donor pocket, Phe177 assumes a conformation that together with complementary shielding from the A-loop prevents the donor

from leaving through a "back door". The conformation of Phe177 is controlled by the precise conformation and position of residues 212–219 in IFH2.

In the donor complex, *Pf*DPMS•GDP-Man•Mn$^{2}$, the side chains of Asp89 and His94 are within hydrogen-bonding distance of the donor mannosyl C2 and C3 hydroxyl groups, respectively (Fig. 2b). The mannosyl O6 and O4 hydroxyl groups are also within hydrogen-bonding distance of the side-chain nitrogen atoms in Arg117 and Lys178. The mannosyl C1 atom is rotated relative the expected point of attack by an incoming nucleophilic Dol-P phosphate group (Fig. 2b), but would be optimally oriented by means of a simple rotation in response to acceptor binding. The *Pf*DPMS•GDP•Mg$^{2+}$ and *Pf*DPMS•GDP-Man•Mn$^{2+}$ complexes are very similar with only minor shifts in side chains located in IFH1 and IFH2, which do not interact directly with GDP or GDP-Man (Fig. 2c). As expected for GT2 members, wild-type *Pf*DPMS activity depends on divalent cations (Supplementary Fig. 6).

**Binding of Dol-*P* acceptor and Dol-*P*-Man product**. The archaeon *P. furiosus* prefers 65-carbon Dol-P carriers[25]. Due to

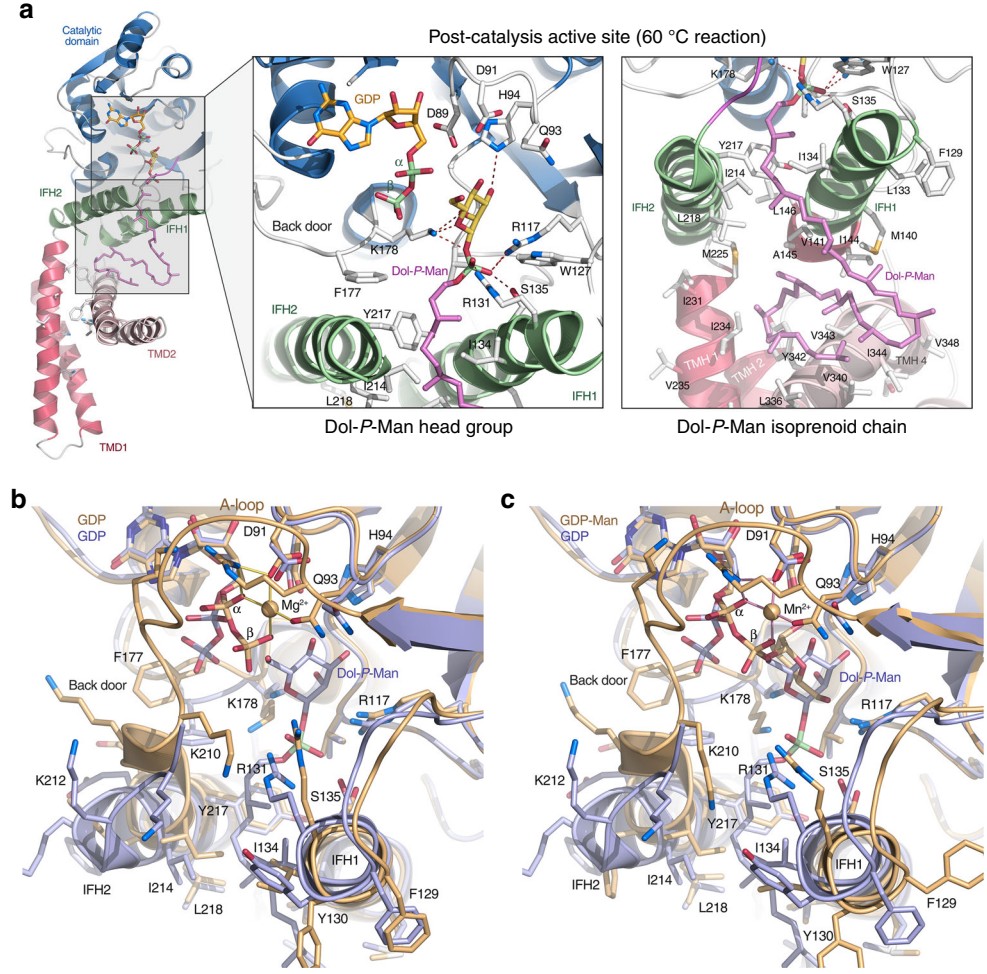

**Fig. 3** Structure of the *Pf*DPMS•GDP•Dol55-*P*-Man complex. **a** Enlarged views of the active site (*left* inset) and the TM domain (*right* inset). *Left* inset, the Dol-*P*-Man head group is shown bound in the active site of wild-type *Pf*DPMS. The metal ion has departed and the GDP diphosphate group has flipped away from the transfer site towards the "back door", to accommodate the glycolipid product. *Right* inset, the Dol-*P*-Man isoprenoid chain traces down to the TM domain where it rests on TMH4. The acceptor loop is open and Phe177 is in the "down" position. Superimposition of the crystal complexes **b** *Pf*DPMS•GDP•Mg$^{2+}$ (*gold*) and *Pf*DPMS•GDP•Dol55-*P*-Man (*blue*), and **c** *Pf*DPMS•GDP-Man•Mn$^{2+}$ (*gold*) and *Pf*DPMS•GDP•Dol55-*P*-Man (*blue*). Relevant side chains are shown

commercial unavailability of 65-carbon Dol-*P*, we tested two alternative carbon-chain lengths, C55 and C95. Both Dol55-*P* and Dol95-*P* serve as acceptors for *Pf*DPMS in vitro (Supplementary Fig. 7a), where Dol95-*P* performs better (37% higher in turnover). Both acceptor lipids provide significant thermal stabilization to unfolding for detergent-solubilized *Pf*DPMS with an increase in melting temperature ($\Delta T_m$) of 13.3 °C and 18.6 °C, respectively, compared with no addition of Dol-*P* (Supplementary Fig. 7b, c). The $\Delta T_m$ value is 40% higher for Dol95-*P* compared with Dol55-*P*, which indicates that the higher activity (37%) with Dol95-*P* is mostly due to protein stabilization. Visualization of the Dol95-*P*-Man product with thin layer chromatography proved challenging due to co-migration of Dol95-*P* and Dol95-*P*-Man in the solvents tested. This problem could be solved by subjecting Dol95-*P*-Man to mild acid hydrolysis using hydrochloric acid, which allowed the mannose released from Dol95-*P*-Man to be unambiguously identified (Supplementary Fig. 8).

In an attempt to capture catalytically relevant structural states of wild-type *Pf*DPMS, we crystallized a reaction mixture that had been incubated at 60 °C prior to crystallization. A crystal grown from this reaction showed a post-catalysis active-site state with both products, GDP and Dol55-*P*-Man, bound (Fig. 3). The

mannosyl O3, O2 and O5 atoms are within hydrogen-bonding distance of His94 Ne2 and Lys178 Nz (Fig. 3, *left* inset). In contrast to the *Pf*DPMS•GDP•Mg$^{2+}$ and *Pf*DPMS•GDP-Man•Mn$^{2+}$ complexes, the diphosphate group of the GDP product has moved away towards the "back door" to provide space for the mannosyl head group of Dol-*P*-Man. To accommodate the new conformation of the GDP diphosphate moiety, Phe177, which gates the back door in the nucleotide and donor complexes, has flipped down to position the aromatic ring immediately below the GDP β-phosphate group (Fig. 3b, c).

The phosphate group in the glycolipid product is coordinated by Ser135, Arg117, and Arg131, and the phosphoester oxygen interacts with Lys178 (Fig. 3, *left* inset). The binding site for the Dol-*P* phosphate group in Dol-*P*-Man shows a distinctly positive electrostatic potential (Supplementary Fig. 9). The importance of Ser135 in IFH1 for binding and positioning the phosphate group in the acceptor lipid is further emphasized by a significant drop in product formation by the S135A variant (Supplementary Fig. 6c), which agrees with the observed effect for an alanine mutant of the equivalent Ser141 in *S. cerevisiae* DPMS[26]. The dolichyl chain is bound between IFH1 and IFH2 where the first two isoprene units interact with hydrophobic residues: Ile134, Ala138, Val141

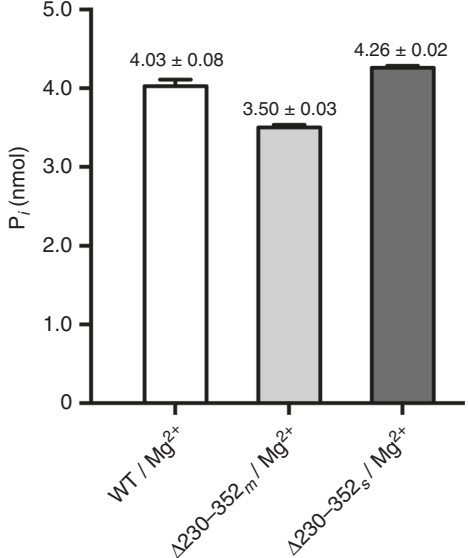

**Fig. 4** Catalytic mechanism of *Pf*DPMS. Catalytic mechanism of the transfer reaction. Nucleophilic attack by the Dol-*P* phosphate oxygen on the mannosyl C1 carbon yields GDP and Dol-*P*-Man. Asp91 and Gln93 coordinate the GDP-Man diphosphate groups via the metal ion, while the key side chains for positioning the Dol-*P* phosphate group for mannosyl transfer are Ser135 and Arg117

**Fig. 5** Activity of *Pf*DPMS wild type and truncation variant Δ230–352. Amount of product formed measured as nanomoles of released free phosphate for wild-type *Pf*DPMS and the truncation variant Δ230–352 lacking the TM domain. Dol55-*P* was used as acceptor substrate. Δ230–352$_m$ and Δ230–352$_s$ were purified from the membrane fraction and aqueous phase, respectively; and they display catalytic activity comparable with that of the wild type. Errors are given as mean values ± s. e.m. (N = 3)

(IFH1) and Ile214, Tyr217, Leu218 (IFH2). Beyond the acceptor-binding site between IFH1 and IFH2, the isoprenoid chain traces down to TMD2, where it folds into a U-shape and rests on TMH4 that acts as a supporting platform (Fig. 3, right inset).

To accommodate binding of the isoprenoid chain, the N-terminal end of IFH2 has changed conformation and shifted away somewhat from IFH1 (Fig. 3b, c). IFH2 is directly connected to the A-loop, and when IFH2 is displaced to accommodate the dolichyl chain, the A-loop is pulled out of the active site to assume a disordered, open conformation. As a consequence of the dislodgement of the A-loop, Arg202 has been expelled from the active site. Moreover, the metal has departed and Gln93 in the $^{89}$DADLQH$^{94}$ motif that used to coordinate the metal ion has been released and points away from the metal-binding site (Fig. 3).

**Catalytic mechanism of *Pf*DPMS.** Based on the *Pf*DPMS crystal complexes, we can outline a catalytic mechanism for Dol-*P*-Man synthesis (Fig. 4). When GDP-Man and metal ion are bound in the active site in the absence of acceptor lipid, the A-loop shields the donor substrate at the "front door", and Phe177 gates the "back door". Based on the post-catalysis GDP•Dol55-*P*-Man product complex, a Dol-*P* molecule would position itself with the phosphate group in the acceptor-phosphate binding site (Ser135, Arg117, and Arg131), with the first two isoprene units of the dolichyl chain between IFH1 and IFH2. The Dol-*P* phosphate group is pre-activated for nucleophilic attack, and not dependent on a catalytic base for activation.

The Dol-*P* phosphate group attacks with one of its activated oxygen atoms at the α-mannosyl C1 and a new bond is formed with inverted configuration at C1[27]. This is in agreement with the mechanism proposed for *S. cerevisiae* DPMS based on theoretical modeling and fluorescence resonance energy transfer experiments[28], however, our present data do not lend support for the suggestion that DPMSs are binuclear metalloenzymes. Only a single metal ion, Mg$^{2+}$ or Mn$^{2+}$, is consistently observed in the *Pf*DPMS complexes, and the high sequence conservation of the active sites in DPMSs suggests that *Pf*DPMS is representative with respect to metal-ion binding stoichiometry. Furthermore, the role of the conserved Arg212 in *Sc*DPMS (Arg202 in the A-loop of *Pf*DPMS) is not, as suggested based on modeling[28], to coordinate the mannosyl O3 atom in GDP-Man and the Dol-*P* phosphate oxygen atom, but to couple metal binding to conformational control of the conserved A-loop. Coordination of the nucleophilic Dol-*P* phosphate group is assigned to Arg117 and Ser135 with assistance of Arg131 (Arg122, Ser141 and Arg137 in *Sc*DPMS), all of which are conserved in bona fide DPMSs (Supplementary Fig. 3).

The involvement of IFH1 and IFH2 in acceptor binding assigns an important role to these helices, which prompted us to investigate the role of the TM domain (TMD1/TMD2) in Dol-*P*-Man synthesis. For this purpose, a truncation variant lacking the

TM domain (residues 230–352) was produced. Interestingly, the Δ230–352 variant was still targeted to the membrane and displayed catalytic activity comparable with that of the wild type (Fig. 5). This strengthens further the role of the IF helices in acceptor recognition and binding, and emphasizes that Dol-*P*-Man synthesis does not per se require the TM domain, and that the topologically variable TM domains of different DPMS classes (Supplementary Fig. 1) perform functions that are distinct from catalyzing Dol-*P* mannosylation.

**Structural comparison with related enzymes**. There are no structures of acceptor-product complexes available for glycosyl-transferases that use lipid substrates, however, crystal structures of donor product complexes (i.e., the donor without sugar) have been reported for two bacterial membrane glycosyltransferases, i.e., *Synechocystis* GtrB (complex with uridine diphosphate, UDP, and Mg$^{2+}$; PDB code 5EKP)[29] and the aminoarabinose transferase ArnT from *Cupriavidus metallidurans* (complex with undecaprenylphosphate, Und-*P*; PDB code 5F15)[30].

GtrB is a GT2 member that catalyzes the transfer of glucose (Glc) from UDP-Glc to Und-*P*[29] in vitro. *Pf*DPMS and GtrB share similar overall topology of the catalytic domain (Supplementary Fig. 2), but differ in the TM domain and IF helices (Supplementary Fig. 10). Compared with *Pf*DPMS, the two TMHs in GtrB assume different positions relative to the catalytic domain, and participate in homotetramer assembly (Supplementary Fig. 10d), and there are no counterparts in GtrB to the TMH-dimer motifs in *Pf*DPMS (Supplementary Fig. 10e). Another topological difference is an extra β-strand that hydrogen bonds with β3 of the GT-A core (Supplementary Figs. 2 and 10a).

Detergent-solubilized *Pf*DPMS is likely to be functional as a monomer, which does not preclude the possibility of higher-order oligomers in a native membrane. To investigate the possibility of higher-order oligomers for *Pf*DPMS, a dimer and tetramer were modeled in silico. Due to the tilted orientation of TMD2, the *Pf*DPMS structure is incompatible with a homotetramer assembly; however, a dimer is possible, albeit predicted not to be stable according to PISA (complex formation significance score value 0.0). Thus, the TM domains in *Pf*DPMS and GtrB probably have different functions. Due to the differences of the IFH and TMH elements (Supplementary Fig. 10e), a structural alignment of *Pf*DPMS and the GtrB subunit is only meaningful for the GT-A domain. Of the 196 and 192 residues that comprise the GT-A domains in *Pf*DPMS and GtrB (PDB code 5EKP), respectively, 146 Cα positions can be aligned with an r.m.s.d. value of ~1.6 Å. Of the aligned positions, 46 amino acids are identical, which corresponds to a structural sequence identity of 31.5% (24% identity for the entire extent of the catalytic domain).

All known bona fide DPMSs use GDP-Man as donor substrate, whereas GtrB use UDP-Glc in vitro. In agreement with the observed activity, UDP is bound in the donor-binding pocket in the GtrB crystal structure. Overall, the inner region of the pocket is similar in the two enzymes (Supplementary Fig. 10f). The $^{89}$DADLQH$^{94}$ motif in *Pf*DPMS corresponds to $^{94}$DADLQD$^{99}$ in GtrB where Asp96 and Gln98 coordinate the metal ion. A number of differences are also noted (Supplementary Fig. 10f). To accommodate binding of the smaller uridine ring, Asp43 in GtrB (Asp39 in *Pf*DPMS) adopts a different conformation to enable a hydrogen bond with the UDP N3 nitrogen atom. In GtrB, the loop connecting β6 and β7 ("back door" loop in *Pf*DPMS) has a different conformation and is folded away from the donor pocket to pack against the C-terminal end of TMH2. The acceptor loop in GtrB (residues 198–204) was not modeled in the crystal structure, which prevents analysis of functionally relevant structural determinants, however, the loop is three residues

shorter and shows a different amino-acid sequence compared with DPMSs (Supplementary Figs. 3 and 10a). To assess the degree of donor-substrate promiscuity in *Pf*DPMS, the catalytic activity was compared for GDP-Man, GDP-Glc and UDP-Glc (Supplementary Fig. 11). The preference for the mannosyl donor is evident, as shown by a four-fold higher product formation with GDP-Man compared with GDP-Glc. Some residual activity is observed also for UDP-Glc, but the value falls within the error estimate of GDP-Man conversion. Spatially, the smaller uridine ring can be accommodated in the guanine-binding site of *Pf*DPMS, and similarly, glucose is compatible with binding at the transfer site. The different interactions with a uridine ring as opposed to guanine, and the presence of an equatorial glucosyl C2 hydroxyl group compared with its axial configuration in mannose, are the main discrepancies that are expected to account for the lower activity with UDP-Glc or GDP-Glc as donor. Equatorial versus axial O2 will mainly affect interactions associated with Asp89, Leu69, and Phe177, all of which appear to have distinct roles in Dol-*P*-Man synthesis.

As for *Pf*DPMS, the IF helices in GtrB are amphipathic, and oriented with the hydrophobic sides facing the membrane, but as noted above, they assume different positions relative the catalytic domain and active site (Supplementary Fig. 10b, c). IFH1 is also shorter in GtrB. The interface between IFH1 and IFH2 in GtrB is densely occupied by large side chains (Phe134, Phe139, Phe178, Met179, Phe183, and Trp213), and packs against a loop segment (residues 150–155) between IFH1 and β6 that blocks a possible opening between the IF helices. A similar block does not exist in *Pf*DPMS where the corresponding loop adopts a different conformation. Unless a major conformation change takes place in the blocking loop and IF helices in GtrB, entry of the Und-*P* acceptor between IFH1 and IFH2 is highly improbable. A more likely entry point for the Und-*P* substrate in GtrB is between IFH2 and the β6–β7 loop, where a large opening can provide direct access to the bound donor substrate. This opening is close to the GtrB A/B (and C/D) subunit interface, and is one of the regions that show pronounced differences in sequence and structure compared with *Pf*DPMS. Because of these discrepancies between the *Pf*DPMS and GtrB structures, it is difficult to identify side chains in GtrB that can serve as counterparts to Arg131 and Ser135 that bind the acceptor Dol-*P* phosphate group in *Pf*DPMS. Arg122 in GtrB probably corresponds to Arg117, which also participates in Dol-*P* phosphate binding in *Pf*DPMS. Furthermore, we predict that Asp99 in the $^{94}$DADLQD$^{99}$ motif of GtrB should be able to interact with the glycosyl O3 hydroxyl group in an Und-*P*-Glc product.

Based on the overall similarity of the substrates and the architecture of the donor-binding pocket, a similar but not necessarily identical, catalytic reaction mechanism can be expected for the two enzymes. In GtrB, Asp157 has been suggested to function as a catalytic acid to protonate the distal β-phosphate group of the donor[29]. This cannot apply to *Pf*DPMS since the corresponding residue is Gly158. A soluble member of the GT2 family, *Staphylococcus aureus* TarS[31] (PDB code 5TZE) has a proline residue in the equivalent position, which suggests that if Asp157 in GtrB plays a role as catalytic acid, it is not a general theme among inverting GT2 enzymes. Because of the absence of acceptor loop in GtrB, and the large differences in the acceptor-binding region, further evaluation of possible similarities and differences with respect catalytic mechanism becomes difficult.

Unlike *Pf*DPMS and GtrB, which are membrane enzymes of the GT2 family, the inverting glycosyltransferase ArnT belongs to family GT83, which features a GT-C fold similar to oligosaccharyl transferases[30]. The catalytic domain is associated with three IF helices (JM1–JM3) and a TM domain composed of 13 TMHs.

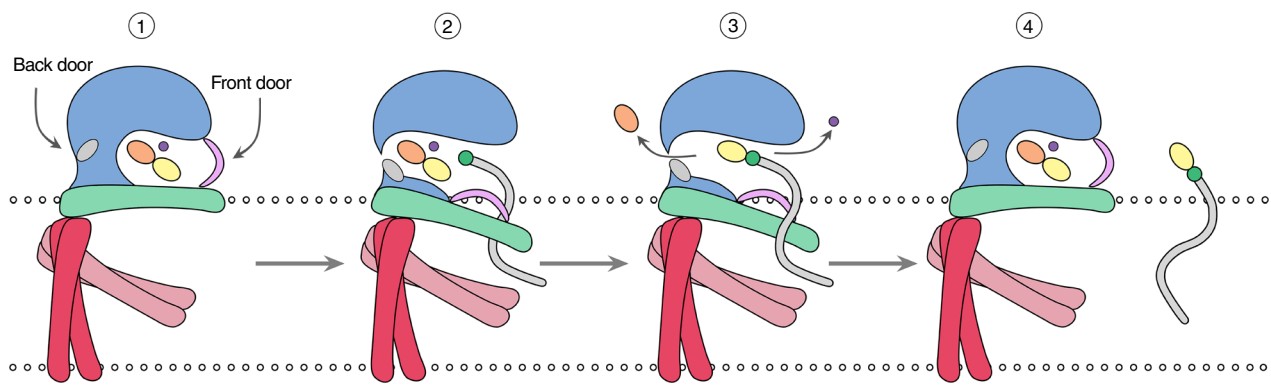

**Fig. 6** Proposed model of substrate binding and product release. Schematic representation of the conformational changes associated with substrate binding and product release. Step 1: prior to encounter with a Dol-P acceptor molecule. GDP-Man and metal ion are bound in the active site. Step 2: Dol-P binding, IFH2 movement, opening of "back door" and of the acceptor loop. Step 3: glycosyl transfer followed and subsequent release of GDP and metal ion. Step 4: binding of GDP-Man and metal ion, release of Dol-P-Man, closure of back door and acceptor loop. See text for further details. Color scheme: Catalytic domain, *blue*; IFH2, *green*; TM domain, *red*; acceptor loop, *pink*; GDP-Man, *orange* and *yellow ovals*; metal ion, *purple circle*; Phe177, *gray oval*; Dol-P isoprenoid chain, *gray*; and Dol-P phosphate group, *green circle*

ArnT does not use a nucleotide sugar as donor substrate, but instead, aminoarabinose is transferred from an undecaprenylphosphate donor to the acceptor substrate lipid A. The structure of ArnT was determined in the apo state (PDB code 5EZM) and in complex with the "unloaded" donor, i.e., Und-P[30] (PDB code 5F15), the latter that would correspond functionally to our GDP•$Mg^{2+}$ complex. The structures and substrate specificities are fundamentally different in $Pf$DPMS and ArnT, but similarities are noted related to the IF helices. As for IFH2 in $Pf$DPMS, the IF helix JM3 in ArnT is directly connected to a flexible loop (A-loop in $Pf$DPMS). In ArnT, this loop is shorter than the A-loop in $Pf$DPMS, and disordered in both ArnT structures. In analogy to IFH2 in $Pf$DPMS, binding of Und-P to ArnT induces a conformational change of JM3 that triggers opening of the acceptor loop, which in turn leads to loss of metal-ion coordination and metal release from the active site. This situation appears similar to the coupling between IFH2 and the metal-binding site in $Pf$DPMS. Beyond this observation, the absence of additional relevant structural states for ArnT precludes further mechanistic comparison of the enzymes.

## Discussion
As observed in the GDP•Dol55-P-Man product complex, several side chains in the IF helices participate in critical interactions with the glycolipid product. This shows that the IF helices, rather than the TM domain, constitute the basis for recognition and binding of the lipid substrate. Although dispensable for activity of detergent-solubilized $Pf$DPMS, the arrangement of TMD1 and TMD2 still offers a binding platform for the Dol-P acceptor and Dol-P-Man product, which is likely to be more relevant for substrate binding and activity in native membranes than in protein-detergent complexes.

Based on the crystal complexes, we can propose a tentative model for substrate binding and product release orchestrated by an interplay between IF helices, donor, metal ion and acceptor. Prior to encounter with a Dol-P acceptor molecule, GDP-Man and metal ion are expected to be bound in the active site (Fig. 2b), with the A-loop secured in its closed conformation by a network of interactions formed between A-loop residues (Arg202 and Lys208), the metal-coordination center (Asp91), and Glu12 in the β1-α1 loop (Fig. 6, step 1). Displacement and conformational changes of the IF helices, most pronouncedly of IFH2, enable

entry of the Dol-P acceptor between IFH1 and IFH2, and docking of the Dol-P phosphate group at Ser135 (assisted by Arg117 and Arg131), which is located immediately below the mannosyl to be transferred. The acceptor-induced changes in the IF helices (Fig. 3) leads to disruption of the A-loop interaction network at the DXD motif and dislodgement of the A-loop, as well as opening of the back door through a conformational change in Phe177 and its immediate environment (Fig. 6, step 2). We predict that collapse of the interaction network and opening of the A-loop coincides with concomitant loss of metal ion and attack by the pre-activated nucleophilic Dol-P oxygen on the mannosyl C1 atom. As observed in the Dol-P-Man complex, the donor product (GDP) moves towards the "back door" to accommodate the mannosyl group (Fig. 3). If GDP would leave prior to release of Dol-P-Man, GDP must exit through the back door, and the metal ion probably departs through the "front door" (Fig. 6, step 3). The cycle is completed by release of products, rearrangement of the IF helices, binding of a new donor molecule and metal ion, and lock-down of the A-loop over the donor-binding pocket (Fig. 6, step 4).

A 13-amino acid sequence in yeast glycosyltransferases, including $Sc$DPMS, was suggested to participate in dolichol recognition[32]. This sequence ($^{246}$LFITFWSILFFYV$^{258}$), referred to the polyisoprenyl-recognition sequence (PIRS), maps to the only existing TMH in $Sc$DPMS, which would correspond topologically to TMH1 in $Pf$DPMS. However, the amino-acid sequences of the two TMHs show weak similarity at best (Supplementary Fig. 12a). In contrast, $Pf$DPMS TMH1 (first TMH in TMD1) and TMH3 (first TMH in TMD2) show more convincing sequence similarity to human DPM2 (Supplementary Fig. 12b, c). While we cannot disprove or confirm the possible importance of the proposed PIRS in $Sc$DPMS, the key determinants of Dol-P recognition can be outlined for $Pf$DPMS, which may also be valid for human DPMS (Supplementary Fig. 12d). In $Pf$DPMS, the amino-acid side chains that are in direct contact with the isoprenoid chain and phosphate group are mainly located in IFH1, IFH2, and TMH4 (Supplementary Fig. 12d). This agrees with the observations that (i) the $Pf$DPMS Δ230–352 variant targets to the membrane of the expression host, and (ii) that the catalytic activity of Δ230–352 variant is similar to that of the full-length enzyme, which emphasizes that dolichyl-chain recognition is intact in the absence of TM domain. In the crystal, the electron density beyond the first two isoprene units is weak, and although

the principal chain trajectory can be discerned, detailed interpretation of interactions between the TM domain and the isoprenoid chain beyond the second IP unit becomes difficult. Moreover, to allow close contacts between *Pf*DPMS molecules in the crystal lattice, the extensive C55 dolichyl chain adopts a U-shape-folded conformation, which is not necessarily the conformation preferred in the membrane environment.

Like other inverting members of the GT2 family, DPMSs (including *Pf*DPMS) depend on metal ions for efficient catalysis[33, 34]. We still observe some residual catalytic activity for the D89A and D91A variants of the DXD motif (Supplementary Fig. 6). Asp89 is not directly involved in metal binding, but appears to be more important for interactions with the mannosyl group. The principal role of the metal ion in *Pf*DPMS is to coordinate and position the diphosphate group of the donor for catalysis. Asp91 and Gln93 coordinate the α- and β-phosphate groups of GDP via the metal ion, respectively (Fig. 2). When the acceptor loop is in the closed state, Arg202 forms ionic interactions with the O2A oxygen atom of the α-phosphate group in GDP and GDP-Man, as well as with Asp91 in the DXD motif and with Glu12 (Fig. 2). This network of ionic interactions provides a means to couple donor binding to conformational control of the A-loop. We predict that Gln93 and Arg202 may be sufficient to rescue some of the metal-binding capacity in D91A to account for the observed residual activity.

The sequences of the *Pf*DPMS catalytic domain and human DPM1 are 39% identical, and both contain the consensus signatures of DPMSs. The close relationship between the enzymes offers a structural basis of understanding human *dpm1*-linked mutations. Dysfunctional human DPM1 is associated with congenital disorder of glycosylation type Ie, which is caused by different mutations associated with the *dpm1* gene (Supplementary Table 2). Of these mutations, three are single amino-acid replacements: R92G, S248P and G152V. The drastic effect of these mutations on DPMS stability and function can be readily inferred from the *Pf*DPMS crystal structures. The conserved Arg63 (Arg92 in *Hs*DPM1) is located at the end of β3 in *Pf*DPMS (Supplementary Figs. 2 and 3) where it helps to secure the guanine-binding loop comprising residues 64–69 through a network of interactions. Insertion of a glycine residue in this position is expected to greatly compromise protein stability, as well as donor binding and catalysis. Glu216 in *Pf*DPMS corresponds to Ser248 in *Hs*DPM1, and is situated in IFH2. In *Pf*DPMS, this region of IFH2, $^{214}$IFEYL$^{218}$ ($^{246}$IVSFL$^{250}$ in *Hs*DPM1), undergoes significant conformational changes in response to lipid binding, and contributes important hydrophobic interactions (underlined) to the Dol-*P* and Dol-*P*-Man isoprenoid chain. These conformational changes regulate, together with Phe177, the coupled opening and closure of the acceptor loop and donor-binding pocket. Insertion of a proline residue in this position, as observed in the human DPM1 mutant S248P, would introduce a kink in IFH2 and compromise conformational responsiveness and acceptor binding, and probably also access of the donor to the active site and/or departure of GDP.

Gly122 (Gly152 in *Hs*DPM1) is conserved in most DPMSs, and is located in the loop between β5 and IFH1. The C$_\alpha$ packs within van der Waals interaction distances of several backbone atoms and introduction of a valine in this position would introduce considerable steric clashes with the backbone regions 116–118 and 196–198. Since Arg117 is responsible (together with Ser135) for positioning the Dol-*P* phosphate group for nucleophilic attack, the G152V mutation is expected to cause severe structural perturbation in the transfer site. An additional four mutations result in premature termination of the *Hs*DPM1 polypeptide chain at various positions, and neither of these gene variants are likely to produce a foldable protein.

Aberrant glycosylation is a hallmark of cancer progression[35]. By analyzing salivary mRNA transcript levels for human *dpm1* and three other genes, pancreatic cancer patients could be differentiated from healthy subjects[36]. In addition, the human *dpm1* gene is one of six glycosylation genes that have been established as significant prognostic markers for breast carcinomas[14]. Angiogenesis is significantly enhanced in malignant breast tumors, and neoplastic transformation is correlated with changes in the glycan pattern on the cell surfaces. cAMP-dependent phosphorylation of Ser141 in *S. cerevisiae* DPMS (and Ser165 in human DPMS[37, 38]) has been observed to increase DPMS activity, and in the case of human DPMS, phosphorylation was associated with a concomitant increase in lipid-linked oligosaccharide, protein N-glycosylation and proliferation of capillary endothelial cells (angiogenesis)[39]. While angiogenesis is a complex process that is influenced by range of factors, our observation that the conserved Ser135 in *Pf*DPMS plays a critical role by positioning the acceptor phosphate group for nucleophilic attack on the mannosyl substrate, rationalizes the importance of this serine for DPMS function. The correlation between DPMS and tumor progression is interesting and highlights DPMS as a possible drug target for development of cancer glycotherapeutics.

## Methods

**Cloning, site-directed mutagenesis and protein production**. The gene coding for DPMS (EC 2.4.1.83) from the euryarchaeon *Pyrococcus furiosus* DSM 3638 (UniProt Q8U4M3) was synthesized and codon optimized for expression in *Escherichia coli* by DNA2.0 (DNA2.0, Inc. Menlo Park, CA, USA). The gene was cloned using ligation-independent cloning[40] into the vector pNIC28-Bsa4[41]. A PCR protocol using 70 ng plasmid DNA and Pfu DNA polymerase (Fermentas, Germany) included 30 cycles of 95 °C, 3 min; 94 °C, 30 s; 65 °C, 8 min; with a final incubation at 65 °C, 10 min. DpnI-digested PCR products were transformed into *E. coli* Mach1 cells (Invitrogen). Site-directed mutagenesis experiments to generate the *Pf*DPMS variants D89A, D91A, S135A and Δ230–352 were carried out using the following forward and reverse primers:5′-3′

D89A_fwd (CGTGTTCGTTGTAATGGCCGCGGGATCTGCAGCATC)
D89A_rev (GATGCTGCAGATCCGCGGCCATTACAACGAACACG)
D91A_fwd (TTGTAATGGATGCGGCCCCTGCAGCATCCGCC)
D91A_rev (GGCGGATGCTGCAGGGCCGCATCCATTACAA)
S135_fwd (CGTAAACTGATCGCCAAGGGCGCAATTATGGTG)
S135_rev (CACCATAATTGCGCCCTTGGCGATCAGTTTACG)
Δ230–352_fwd (TACTTCCAATCCATGAAAGTAAGCGTCATCATCCCAACC)
Δ230–352_rev (TATCCACCTTTACTGTCAACCTTCCCATTTCATCAGACGG)

The PCR reactions were carried out using 50 ng plasmid DNA and Phusion High-Fidelity DNA polymerase (Thermo Fisher), and the resulting PCR products were transformed into *E. coli* DH5α cells grown on LB agar supplemented with 50 µg ml$^{-1}$ kanamycin.

The recombinant plasmids were transformed into *E. coli* C41 (DE3) competent cells. Cultures containing Terrific Broth medium supplemented with kanamycin were induced with β-D-1-thiogalactopyranoside and incubated at 17 °C. The cells were harvested, and the bacterial pellet resuspended in 25 mM K$_2$HPO$_4$ (pH 7.2), 150 mM NaCl, 5% (v/v) glycerol, with complete protease inhibitor cocktail (Roche). The resuspended pellet was homogenized using an AVESTIN Emulsiflex-C3 system, followed by centrifugation at 10,000 r.p.m. (Beckman Coulter JA-25.50 fixed-angle rotor, 12,096×g) with an Avanti J-20XP centrifuge (Beckman Coulter) for 10 min at 4 °C. The resulting supernatant was centrifuged using an Optima LE-80K centrifuge (Beckman Coulter 45 Ti rotor) at 35,000 r.p.m. (125,749×g) for 70 min (4 °C) to pellet the membrane fraction. The membrane fractions were resuspended in 25 mM K$_2$HPO$_4$ (pH 7.2), 150 mM NaCl, 5% (v/v) glycerol with a tissue grinder, flash frozen in liquid nitrogen and stored at −80 °C.

**Protein purification**. Membrane proteins were solubilized from membrane fractions in the presence of 1% n-dodecyl-β-D-maltoside (DDM, Anatrace) in 25 mM K$_2$HPO$_4$ (pH 7.2), 150 mM NaCl and 5% (v/v) glycerol at 4 °C. Extracted membrane proteins were recovered by centrifugation at 30,000 r.p.m. (92,387×g) at 4 °C (45 Ti rotor, Optima E-80K centrifuge; Beckman Coulter). The supernatant was mixed with Ni-NTA agarose resin (Invitrogen) and packed in Bio-Rad Econo-Pac columns, and left to incubate at 4 °C for 1 h. The columns were washed with 25 mM K$_2$HPO$_4$ (pH 7.2), 150 mM NaCl, 30–50 mM imidazole, 5–10% (v/v) glycerol, containing 0.05–0.1% DDM or 0.05% N,N-dimethyldodecylamine N-oxide (LDAO, Anatrace). Bound protein was eluted with buffer containing 500 mM imidazole. Protein samples were treated with 10 mM ethylenediamine tetraacetic acid and 10 mM ethyleneglycol tetraacetic acid, concentrated using Vivaspin20 centrifugal concentrators (polyethersulfone filter, molecular weight cut-off, MWCO, 50 kDa), and loaded onto a HiLoad 16/60 Superdex 200 prep grade

column (GE Healthcare Life Sciences) equilibrated with 50 mM 4-(2-hydro-xyethyl)-1-piperazineethanesulfonic acid (HEPES) pH 7.5, 150 mM NaCl, 5–10% (v/v) glycerol, and 0.02% DDM or 0.05% LDAO. The recovered fractions containing PfDPMS were pooled and further concentrated to 15–20 mg ml$^{-1}$ using a Vivaspin20 concentrator (MWCO 50 or 100 kDa). The PfDPMS truncation variant Δ230–352 includes only the catalytic domain, but because of the hydrophobic IF helices, 80% of the protein targets to the membrane, while the remaining 20% fraction appears as a water-soluble fraction. Therefore, variant Δ230–352 was prepared as two fractions, one purified from the membrane fraction (Δ230–352 $_m$) and one from the aqueous phase (Δ230–352 $_s$).

**Structure determination and model refinement**. LDAO-solubilized PfDPMS was used for crystallization. All crystals were grown by vapor diffusion in sitting drops at 4 °C in the presence of metal ions and substrates (Supplementary Table 1). For the GDP•Mg$^{2+}$ and GDP-Man•Mn$^{2+}$ structures, the protein was pre-mixed with 5 mM metal ion (MgCl$_2$ or MnCl$_2$) and either 5 mM GDP or GDP-Man in 50 mM HEPES (pH 7.5), 150 mM NaCl, 10% (v/v) glycerol, and 0.05% LDAO. Successful crystallization was achieved in the presence of 0.2 M potassium chloride, 0.1 M trisodium citrate (pH 5.5), and 37% (v/v) pentaerythritol propoxylate (5/4 PO/OH).

The GDP•Dol55-P-Man structure was obtained from a reaction mixture containing PfDPMS, GDP-Man, Mn$^{2+}$ and Dol55-P incubated at 60 °C. The reaction mixture was allowed to cool down after which it was dispensed in crystallization plates. Dol55-P was selected for crystallization to avoid obstruction of the crystal packing by longer dolichol chains. For experimental phasing, crystals were subjected to heavy atoms (PbCl$_2$, K$_2$PtCl$_4$ or merthiolate) at a final concentration of 3–5 mM.

Intensity data for non-derivatized and derivatized crystals were recorded at synchrotron facilities (Supplementary Table 1), followed by merging and scaling using the XDS package[42]. The crystals belong to space group C222$_1$ with one molecule in the asymmetric unit and an approximate solvent content of 53%. Heavy-atom substructure determination and phasing were performed by multiple isomorphous replacement with anomalous scattering (MIRAS) using the program autoSHARP[43] included in the SHARP package[44] and four heavy-atom derivatives (Supplementary Table 1). Fourier amplitudes for the GDP•Mg$^{2+}$ crystal were used as native. The single platinum site in derivative 2 was kept although its contribution to phasing is minor. The resulting electron density was improved and phases extended to 2.0 Å by solvent flattening using SOLOMON[45] as implemented in SHARP[44].

An initial model for the GDP•Mg$^{2+}$ structure was built by alternating manual model building using COOT[46] and program O[47], and refinement with PHENIX[48] guided by likelihood-weighted electron-density maps. The Fourier amplitudes for the GDP-Man•Mn$^{2+}$ and GDP•Dol55-P complexes were phased using Fourier synthesis with model phases from the refined GDP•Mg$^{2+}$ structure. Refinement with PHENIX included refinement of XYZ coordinates, real-space refinement, and refinement of individual atomic displacement parameters. Refinement statistics are given in Supplementary Table 1, and unbiased electron density for the ligands in the individual models are shown in (Supplementary Fig. 13).

**Activity assays**. For the glycosyltransferase (GT) reaction, PfDPMS (0.014 mM enzyme in 50 mM HEPES pH 7.5, 150 mM NaCl, 5% (v/v) glycerol, 0.02% DDM or 0.05% LDAO) was incubated at 75 °C in a volume of 45 μl containing 0.23 mM donor substrate (GDP-Man, GDP-Glc, UDP-Glc; Sigma-Aldrich), 0.25 mM Dol55-P (C55-dolichyl monophosphate; Larodan, Solna, Sweden; Cat. No. 67-1055) or 0.16 mM Dol95-P (C95-dolichyl monophosphate; Larodan, Solna, Sweden; Cat. No. 67-1095), 10 mM MgCl$_2$ (or CaCl$_2$, or MnCl$_2$). To quantify conversion, we used a protocol based on the universal colorimetric phosphate-coupled glycosyltransferase assay (R&D Systems, Inc., Minneapolis, MN, USA)[49, 50]. After completion of the GT reaction, the samples were cooled and ultrafiltrated using Amicon Ultra filters (NMWL 10 kDa; Millipore) to remove protein and minimize the detergent content. Aliquots of 15 μl ultrafiltrate were diluted five-fold by adding 1X assay buffer (25 mM Tris pH 7.5, 10 mM CaCl$_2$, 10 mM MnCl$_2$). A 5-μl aliquot containing 10 ng μl$^{-1}$ recombinant human CD39L3/ENTPD3 protein (ectonucleoside triphosphate diphosphohydrolase) in 4x assay buffer was added to 45 μl of the diluted ultrafiltrate sample, and incubated at 25 °C for 16 h. Each sample was then further diluted with H$_2$O to a final volume of 100 μl and transferred to a 96-well plate (Greiner). A volume of 25 μl malachite green development solution (10 parts malachite green stock solution, 2.5 parts 7.5% ammonium molybdate, and 0.2 parts 11% Tween 20) was added to each sample. The samples were incubated at room temperature for 15 min, after which the OD at 630 nm was read (Omega Fluostar; BMG Labtech). To assess the metal dependency of wild-type PfDPMS, the activity was measured in reactions without added metal ion, but in the presence of ethyleneglycol tetraacetic acid and ethylenediamine tetraacetic acid (final concentrations 1, 10, and 25 mM). The activity assay was otherwise performed as described above. The data were analyzed with GraphPad Prism 7 (GraphPad Software, San Diego California USA).

**Modeling of Dol55-P-Man**. The acceptor substrate Dol55-P used for crystallization contains 11 isoprene units with the assignment ω-t$_2$-c$_7$-S-P; where the ω-terminal isoprene is unsaturated and in trans configuration, followed by two

internal trans-configured (t) isoprene units and seven internal cis-configured (c) isoprene units. As in archaeal and eukaryal dolichols, the α-terminal isoprene unit (attached to the phosphate group) is saturated (S) and in cis configuration. The same isoprene assignment applies to the Dol95-P, but the polyisoprenoid chain is longer, ω-t$_2$-c$_{15}$-S-P. Pyrococcus furiosus uses mainly a 65-carbon dolichyl monophosphate with 13 isoprene units corresponding to ω-t$_2$-c$_9$-S-P[25]. Both the α- and ω-terminal isoprene units are saturated, as well as at least six of the nine internal cis-configured isoprene units. Thus, except for the precise polyisoprenoid chain length and the saturation of the ω-terminal isoprene, our Dol55-P substrate agrees well the natural acceptor substrate. The coordinates for C15-dolichyl monophosphate were retrieved from PubChem (http://pubchem.ncbi.nlm.nih.gov) and extended by manual editing using molecular graphics (COOT) by eight isoprene units to a 55-carbon chain. The dihedral angles were adjusted and the resulting Dol55-P model energy minimized using the YASARA server (http://www.yasara.org/minimizationserver.htm)[51]. The minimized coordinates were close to the starting coordinates. A separate model was produced by adding a β-linked mannosyl group to the terminal phosphate group. The Dol55-P-Man model was then fitted according to the experimental electron density for the post-catalysis structure.

**Thermal shift assay**. To analyze thermal stability to unfolding, samples containing 10 μg protein were incubated for 10 min at temperatures between 45 and 100 °C in the absence or presence of Dol55-P or Dol95-P using a PCR thermal cycler (Veriti 96-Well Thermal Cycler with VeriFlex blocks, Applied Biosystems). Following heat treatment, the samples were cooled and precipitated protein material was removed by centrifuging at 14,500 r.p.m. for 10 min at 4 °C. Supernatants, containing protein that remain soluble, were analyzed by Coomassie-stained SDS-PAGE. The gels were scanned and densitometric analysis was performed with ImageJ (Rasband, W. S., ImageJ, U. S. National Institutes of Health, Bethesda, Maryland, USA, http://imagej.nih.gov/ij/, 1997–2016). The data were fitted with Prism GraphPad Prism 7 (GraphPad Software, San Diego California USA).

**Thin-layer chromatography**. Reaction mixtures contained 10 nmol PfDPMS (in 50 mM HEPES pH 7.5, 150 mM NaCl, 5% (v/v) glycerol, 0.07% LAPAO), 400 nmol GDP-Man, 40 nmol Dol95-P (dissolved in 50 mM HEPES pH 7.5, 150 mM NaCl, 0.07% LAPAO), and 750 nmol MnCl2 in 50 mM HEPES (pH 7.5) with 150 mM NaCl, and 0.07% LAPAO. Once the reaction was run to completion, the samples were extracted with chloroform:methanol [1:1] and dried using a SpeedVac (model Savant AES2010). To release the mannose head group, the apolar phase (sample 2 in Supplementary Fig. 8) containing the Dol95-P-Man product was subjected to mild acid hydrolysis as follows: the apolar phase sample was re-dissolved in 50% n-propanol and 50 mM HCl (in 50% n-propanol) was added, followed by incubation of the sample at 95 °C for 30 min. The acid-treated sample was extracted to separate the mannose-containing polar phase using chloroform, and the apolar phase was washed twice with water. A reaction mixture without added PfDPMS and one without Dol95-P were treated identically and used as negative control. For all reaction samples, both apolar and polar phases were analyzed by thin-layer chromatography. Thin-layer chromatography was performed on glass HPTLC Silica Gel 60 plate (Merck, Millipore) in chloroform:methanol:water (65:25:4, v-v: v), and stained with a solution of p-anisaldehyde in ethanol and sulfuric acid.

**Data availability**. Atomic coordinates and structure factor amplitudes have been deposited with the Protein Data Bank (www.rcsb.org) under accession codes 5MLZ (PfDPMS•GDP•Mg$^{2+}$), 5MM0 (PfDPMS•GDP-Man•Mn$^{2+}$) and 5MM1 (PfDPMS•GDP•Dol55-P-Man). The data that support the findings of this study are available from the corresponding author upon request.

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

## Acknowledgements

We thank the beamline staff scientists for support during data collection at beamlines PROXIMA 1 at SOLEIL (France) and I03 at Diamond Light Source (UK). We acknowledge financial support to C.D. from the Swedish Research Councils Formas (grants no 2012–915) and VR (grants no's 2011–5768 and 2013-5717). The research leading to these results has received funding from the European Community's Seventh Framework Programme (FP7/2007–2013) under BioStruct-X (grant agreement no 283570). Part of this work was facilitated by the Protein Science Facility at Karolinska Institutet/SciLifeLab (http://psf.ki.se).

## Author contributions

T.-C.T. performed the original cloning and expression of the *P. furiosus dpms* gene; T.R., T.-C.T., and R.G. performed the mutagenesis experiments; R.G. performed gene expression, protein purification, protein characterization, protein crystallization, heavy-atom screening, and synchrotron data collection; C.D. performed heavy-atom phasing, structure determination, and supervised the work; R.G. and C.D. performed model refinement and analysis of structural and biochemical data; R.G. and C.D. designed the experiments and analyzed the data. R.G., T.R, T.-C.T., and C.D. wrote the manuscript.

## Additional information

**Competing interests::** The authors declare no competing financial interests.

