## [Peer Review File · Nature Communications]

PEER REVIEW FILE

Reviewers' Comments:

Reviewer #1 (Remarks to the Author):

“Structural basis of dolichylphosphate mannose synthesis” by Gandini and colleagues represents a significant advance in our understanding of dolichylphosphate mannose synthases (DPMS) at the molecular level. Specifically, the authors report the crystal structures of the DPMS from *Pyrococcus furiosus* (PfDPMS) in complex with (i) GDP, (ii) GDP-Man and (iii) GDP and Man-Dol55-P. Based on structural and mutagenesis data a molecular mechanism for substrate binding/product release and catalysis is proposed. Because of its novelty the manuscript is therefore suitable for publication in Nature Communications. However, the manuscript/interpretation of the experimental data can be greatly improved:

1. Please consider to replace “Structural basis of dolichylphosphate mannose synthesis” by “Structural basis of dolichylphosphate mannose biosynthesis”

2. lines 8 to 18: Please reduce/modify the introduction. The reaction catalyzed by DPMS should be clearly stated in the abstract. In addition, I would strongly suggest to increase the description of your findings: e.g. (i) protein architecture, (ii) protein complexes and (iii) catalytic mechanism.

3. line 16: replace “repsonsible” by “responsible”

4. lines 26 to 28: “Dol-P-Man, is the only source of activated mannose for protein glycosylation and GPI anchor synthesis in the endoplasmic reticulum of eukaryotes”. GDP-Man is utilized by several GTs localized in the cytoplasmic phase of the ER membrane.

5. lines 42 to 43: “Based on differences in the TM domain, two classes of DPMS have been proposed 17”. Please introduce a panel showing Type-I enzymes topology in Fig. S2.

6. lines 60 to 61: “Firstly, we determined the crystal structures of PfDPMS in complex with metal-bound nucleotide (GDP•Mg²⁺) and donor (GDP-Man•Mn²⁺)”. The GDP•Mg²⁺ complex could/might be considered as a complex of PfDPMS with nucleotide-metal product.

7. lines 66 to 67: “The catalytic domain belongs to family 2 of inverting glycosyltransferases (www.cazy.org)”. Please state that the catalytic domain displays a GT-A fold. It would be useful

to include a topology diagram as Fig. S2b.

8. line 68: Are IFH1 and IFH2 amphipathic helices? What is the electrostatic surface potential of PfDPMS? A good opportunity to include/discuss this information in the "Results" section and include a new panel in Figure 2 and 3 - please include the location of the substrates.

9. lines 69 to 71: "TMD1 is perpendicular to the membrane and spans 45 Å (Fig. 1a), while TMD2 is rotated by 60° with respect to TMD1, giving the impression of walking legs that associate at a junction." The conformation of TMD1 and TMD2 is functional or a result of packing interactions? Please add a Figure in SI Section, showing the packing interactions. Would the authors expect a similar conformation for the TMD1/TMD2 helices in the context of the membrane? Any sign of – functional - protein oligomerization? Please discuss all these points in the "Results" section.

9. lines 79 to 80: Please describe the location of the GDP-Man binding site in the framework of a GT-A domain. Which secondary structure elements define the GDP-Man binding pocket? A good opportunity to include this information as a new panel in Figure 2.

10. lines 86 to 88: "Wild-type activity is enhanced in the presence of divalent cations, but is not strictly metal-ion dependent (Fig. 2c), as alanine replacements of Asp89 and Asp91 retain activity in both presence and absence of metal ion (Fig. 2d)." Did the authors measure the enzymatic activity in the presence of EDTA?

11. In addition to "Structure of PfDPMS in complex with donor substrate" and "Structure of PfDPMS in complex with glycolipid product", I would suggest to include another section describing the conformational changes observed between the three crystal structures. Please, prepare several figures accordingly. Finally, taking into account (i) the architecture of PfDPMS and (ii) the absence of a membrane, the authors should be very cautious with the interpretation of the conformational changes observed and the impact in the proposed model.

For example:

- lines 102 to 104: "A crystal grown from the 60°C reaction showed a post-catalysis active-site state with both products, GDP and Dol55-P-Man, bound (Fig. 3)." Please superimpose the (i) GDP and (ii) GDP/Man-Dol55-P complexes, showing the conformational differences of GDP and discuss accordingly.

- lines 115 to 116: "The opening of the acceptor loop in response to acceptor binding leads to an increased separation of the two IF helices." Please superimpose the (i) GDP-Man and (ii) GDP/Man-Dol55-P complexes, showing the conformational changes. Please prepare Figures and

discuss them accordingly.

- I would also suggest to prepare a separate section focused in the catalytic mechanism of PfDPMS.

12. lines 123 to 123: “Mechanism of DoI-P-Man synthesis. Based on the PfDPMS crystal complexes, we can outline a detailed mechanism for DoI-P-Man synthesis where substrate binding and concomitant movements of the IF helices control conformational states that are necessary for productive outcome”. What is the experimental evidence that support the substrate binding/product release order? In the absence of experimental enzymatic/kinetic data, the authors should be very cautious with the interpretation of the structural data. In that sense, I strongly encourage to move this entire section to the “Discussion” section.

13. lines 175 to 176: please avoid “the first-in-class” and “the first”.

14. lines 182 to 192: I would suggest to move this paragraph to SI.

15. Similarly, lines 193 to 194: please avoid “There are no structures of acceptor-product complexes available for any glycosyltransferases that use lipid substrates.” Instead, I would suggest to deeply discuss the impact of the PfDPMS-GDP-Man-DoI55-P complex in the understanding of other members of the GT2 family and/or other GTs.

16. lines 195 to 196: “however these structures are too different from true DPMSs to make a structural comparison meaningful.” I still suggest the authors to discuss more in deep and extensively the structural superposition/homology/differences of GtrB and PfDPMS in the context of enzymatic activity and fold – (i) r.m.s.d. value for equivalent residues of each protein, (ii) location of the nucleotide sugar binding site in the GT-A domain, (iii) donor substrate specificity, (iv) importance of the juxtamembrane helices in both enzymes, (v) catalytic mechanism.

17. Figure 4 should be divided as two different Figures: one referred to the catalytic mechanism, and a second one referred to the proposed model of substrate binding and product release – in the absence of experimental data supporting it – in the context of the “Discussion” section.

18. Supplementary Table 1

- The following information should be introduced as first row in the Table: 5MLZ (PfDPMS•GDP•Mg²⁺), 5MM0 (PfDPMS•GDP-Man•Mn²⁺), and 5MM1 (PfDPMS•GDP•DoI55-P-Man).

I support publication of the manuscript in the case all these questions can be addressed satisfactorily.

Reviewer #2 (Remarks to the Author):

Gandini and colleagues have determined structures of the dolichylphosphate β -D-mannose synthase from *P. furiosus*, in an empty unbound and with bound donor and acceptor substrates. In humans glycosylation is an important posttranslational modification, and aberrant glycosylation often has severe pathological consequences. The *P. furiosus* enzyme is predicted to be topologically similar to the human enzyme (on which no 3D-structural information is available), and, therefore, the results are also of great importance to understanding human glycosylation pathologies.

The research has been done competently. The crystals used have high resolution (2.0 Å resolution for the empty enzyme), and thus the results appear solid.

The description of the research, the results, and their implications is clear, as are the figures

Only one part is not very clear to me, and that is the experiment at 60 °C to trap the product. What was the rationale? The methods are not very clear about whether only this experiment was done at 60 °C, or all crystallization experiments. Was crystallization at ambient temperatures not successful? Why was crystallization at 60 °C done, rather than doing it more slowly at room temperature? Do the authors have data to support this approach? I am aware that catalysis is always better at higher temperatures because a larger fraction of the substrate molecules can attain the energy level of the transition state. But crystallization is such lengthy experiment that I could imagine that temperature does not matter that much.

Responses to Reviewers' Comments:

Reviewer #1 (Remarks to the Author):

1. Please consider to replace “Structural basis of dolichylphosphate mannose synthesis” by “Structural basis of dolichylphosphate mannose biosynthesis”

→ **Author comments:** we have updated the title according to the suggestion. We also realized the use of “of” was incorrect in this context and replaced it by “for”.

2. lines 8 to 18:

Please reduce/modify the introduction. The reaction catalyzed by DPMS should be clearly stated in the abstract. In addition, I would strongly suggest to increase the description of your findings: e.g. (i) protein architecture, (ii) protein complexes, (iii) catalytic mechanism.

→ **Author comments:** We have updated the abstract and introduction. However, since the abstract should not exceed 150 words, a more thorough description of architecture, complexes and mechanism is given in the main text.

3. line 16: replace “reponsible” by “responsible”

→ **Author comments:** The spelling error has been corrected.

4. lines 26 to 28: “Dol-P-Man, is the only source of activated mannose for protein glycosylation and GPI anchor synthesis in the endoplasmic reticulum of eukaryotes”. GDP-Man is utilized by several GTs localized in the cytoplasmic phase of the ER membrane.

→ **Author comments:** We were referring to lipid-activated mannose for use at the luminal side of ER, and realize that this caused confusion since it was accidentally left out in the sentence. The sentence has been rewritten to clearly state that we refer to biogenic membranes (ER and prokaryotic cytoplasmic membranes). See p.1 (1st paragraph) in the revised manuscript.

5. lines 42 to 43: “Based on differences in the TM domain, two classes of DPMS have been proposed 17”. Please introduce a panel showing Type-I enzymes topology in Fig. S2.

→ **Author comments:** The information has been added in Fig. S1 of the revised manuscript.

6. lines 60 to 61: “Firstly, we determined the crystal structures of PfDPMS in complex with metalbound nucleotide (GDP•Mg²⁺) and donor (GDP-Man•Mn²⁺)”. The GDP•Mg²⁺ complex could/might be considered as a complex of PfDPMS with nucleotide-metal product.

→ **Author comments:** Yes, however, since the enzyme used for crystallization in the case of the GDP•Mg²⁺ complex had never been exposed to acceptor substrate, we cannot be sure that a GDP•Mg²⁺ complex exists after catalysis. We could see no sign of metal ion in the GDP•Dol-*P*-Man product complex, which raises the possibility that the metal ion departs before the products (GDP and Dol-*P*-Man) are released, which is why we refrained from referring to the GDP•Mg²⁺ complex a “nucleotide-metal product” complex. But principally, yes, it is a possibility if GDP remains, or rebinds, after the Dol-*P*-Man has departed.

7. lines 66 to 67: “The catalytic domain belongs to family 2 of inverting glycosyltransferases (www.cazy.org)”. Please state that the catalytic domain displays a GT-A fold. It would be useful to include a topology diagram as Fig. S2b.

→ **Author comments:** The information about the GT-A fold has been added in the text as well as in a new supplementary figure (Fig. S2) that shows topology diagrams for a selection of GT” members: two membrane and two soluble GT2s.

8. line 68: Are IFH1 and IFH2 amphipathic helices?

→ **Author comments:** Yes, the IF helices are amphipathic. We have included an additional picture in SI, as Fig. S4, to show the amphipathicity.

What is the electrostatic surface potential of PfDPMS? A good opportunity to include/discuss this information in the "Results" section and include a new panel in Figure 2 and 3 - please include the location of the substrates.

→ **Author comments:** Electrostatic surfaces have been inserted in SI as Fig. S9, and is discussed on p. 10, 2nd paragraph.

9. lines 69 to 71: “TMD1 is perpendicular to the membrane and spans 45 Å (Fig. 1a), while TMD2 is rotated by 60° with respect to TMD1, giving the impression of walking legs that associate at a junction.” The conformation of TMD1 and TMD2 is functional or a result of packing interactions? Please add a Figure in SI Section, showing the packing interactions. Would the authors expect a similar conformation for the TMD1/TMD2 helices in the context of the membrane?

→ **Author comments:** We have included an analysis in the Results section on p. 6 (2nd paragraph) relating to the unusual tilt of TMD2 (relative TMD1), and added a new figure

in SI (Fig. S5) showing the packing and interactions. As detailed in the text, a number of observations argue for an authentic arrangement of TMD1 and TMD2.

[REDACTED]

[REDACTED]

[REDACTED]

9. lines 69 to 71 cont: Any sign of – functional - protein oligomerization? Please discuss all these points in the “Results” section.

→ **Author comments:** In the crystal, crystallographic copies of detergent-solubilized *PfDPMS* molecules are arranged such that two unique interfaces are formed between the TM domains. PISA analysis dismisses both interfaces “as non-stable”. Furthermore, there are detergent molecules packing between the TM domains of individual crystallographically related molecules, which together with results from PISA analyses strongly argue against a natural higher-order oligomer for detergent-solubilized *PfDPMS* in the crystal. We have also modeled higher-order oligomers. Because of the tilted TMD2,

a homotetramer is not possible. A dimer is however possible, but the interface is not considered stable in solution (as judged by PISA). The crystal packing is discussed on p. 6 (2nd paragraph) and shown in Fig. S5 (see previous point), and the oligomerization is discussed on p. 13 (3rd paragraph) in the revised manuscript as part of the structural comparison with GtrB.

As a complementary note, we also performed time-dependent cross-linking using glutaraldehyde followed by SDS-PAGE analysis (see picture below). We used the natural tetrameric pyranose 2-oxidase as control. As can be expected for a natural homotetramer, the control protein passes through four distinct steps towards the final homotetrameric state. In contrast, LDAO-solubilized *Pj*DPMS (monomeric molecular weight of 40 kDa) produces a “smear” typical of non-specific crosslinking. There is maybe a weak tendency to form dimers, but in our experience the tentative dimer band is much too diffuse to indicate a natural dimer, especially since the protein undergoes further non-specific crosslinking. This strengthens the conclusion above that *Pj*DPMS behaves mainly as a monomer when solubilized in LDAO.

We conclude that, at present, the available data support that *Pj*DPMS is monomeric in protein-detergent complexes, but we cannot rule out higher-order oligomers in native pyrococcal membranes. *Pj*DPMS originates from a hyperthermophilic archaeon with a membrane that, by the incorporation of tetraether and diether lipids, is adapted to tolerate high temperatures. This should produce a somewhat different membrane environment compared with the membrane of the expression host (*E. coli*). The point being made is that regardless of what oligomeric state *Pj*DPMS may prefer in detergent or in the *E. coli* membrane, it may still not be valid for the pyrococcal membrane, and the only way to arrive at a definitive answer would be to analyze the oligomeric state of the enzyme in the *P. furiosus* membrane.

10. lines 79 to 80: Please describe the location of the GDP-Man binding site in the framework of a GT-A domain. Which secondary structure elements define the GDP-Man binding pocket? A good opportunity to include this information as a new panel in Figure 2.

→ **Author comments:** As mentioned above, we have included a new figure in SI (Fig. S2) that shows the GT-A fold topology for a selection of GT2 members. In addition, the architecture of the active site has been described with reference to the GT-A fold in the text (see p. 7 - Binding of donor substrate).

11. lines 86 to 88: “Wild-type activity is enhanced in the presence of divalent cations, but is not strictly metal-ion dependent (Fig. 2c), as alanine replacements of Asp89 and Asp91 retain activity in both presence and absence of metal ion (Fig. 2d).” Did the authors measure the enzymatic activity in the presence of EDTA?

→ **Author comments:** The control experiment where EGTA and EDTA are added to the reaction mixtures is shown in the updated Fig. S6a. For all samples, the protein was pre-treated with 1 mM EGTA and 1 mM EDTA during the purification procedure. In the graph, the first two bars show the activity for *PfDPMS* (0.014 mM) in the presence and absence of 10 mM Mg^{2+} , respectively (without any addition of chelator). The next three bars show the activity of *PfDPMS* (0.014 mM) without added metal and with increasing final concentrations of EGTA/EDTA (i.e. 1, 10 and 25 mM of each chelating agent). Beyond 10 mM EGTA+10 mM EDTA, no further decrease in activity is observed. The remaining residual activity at this point is about 13% of wild-type activity in the presence of metal ion. Thus, it seems that: either it is not possible to chelate all metal ion present even at high excess of EGTA/EDTA (possibly the metal is protected by the enzyme?); or the enzyme retains some functionality in the absence of metal.

12. In addition to “Structure of *PfDPMS* in complex with donor substrate” and “Structure of *PfDPMS* in complex with glycolipid product”, I would suggest to include another section describing the conformational changes observed between the three crystal structures. Please, prepare several figures accordingly.

→ **Author comments:** We have moved the more “speculative” aspects of discussion regarding substrate binding and product release to the Discussion section (p. 18), and also added new panels to Figs 2 and 3 with overlays of the different complexes.

Finally, taking into account (i) the architecture of *PfDPMS* and (ii) the absence of a membrane, the authors should be very cautious with the interpretation of the conformational changes observed and the impact in the proposed model.

→ **Author comments:** We have tried to avoid overinterpretation, and rewritten the text in the Results to focus more on the features that we consider likely to be “authentic”, i.e. within reasonable probability likely to exist also in the membrane state.

For example:

- lines 102 to 104: “A crystal grown from the 60°C reaction showed a post-catalysis active-site state with both products, GDP and Dol55-P-Man, bound (Fig. 3).” Please superimpose the (i) GDP and (ii) GDP/ Man-Dol55-P complexes, showing the conformational differences of GDP and discuss accordingly.

→ **Author comments:** As mentioned above, new panels showing superimposed complexes have been added as new panels – Fig. 2c (GDP and GDP-Man complexes), Fig. 3b (GDP and GDP/Dol-P-Man complexes), and Fig. 3c (GDP-Man and GDP/Dol-P-Man complexes).

- lines 115 to 116: “The opening of the acceptor loop in response to acceptor binding leads to an increased separation of the two IF helices.” Please superimpose the (i) GDP-Man and (ii) GDP/Man- Dol55-P complexes, showing the conformational changes. Please prepare Figures and discuss them accordingly.

→ **Author comments:** covered in the response to the previous comment.

I would also suggest to prepare a separate section focused in the catalytic mechanism of PfdPMS.

→ **Author comments:** We renamed the section “Mechanism of Dol-P-Man synthesis” in the Results to “Catalytic mechanism of PfdPMS”. To accommodate this point, we have also moved the discussion about conformational changes to the Discussion section as outlined in point 12 above. As a consequence, figures have been split and renumbered.

13. lines 123 to 123: “Mechanism of Dol-P-Man synthesis. Based on the PfdPMS crystal complexes, we can outline a detailed mechanism for Dol-P-Man synthesis where substrate binding and concomitant movements of the IF helices control conformational states that are necessary for productive outcome”. What is the experimental evidence that support the substrate binding/product release order? In the absence of experimental enzymatic/kinetic data, the authors should be very cautious with the interpretation of the structural data. In that sense, I strongly encourage to move this entire section to the “Discussion” section.

→ **Author comments:** We fully agree. Since this part is more speculative, it has been moved to the Discussion as detailed in point 12 above.

14. lines 175 to 176: please avoid “the first-in-class” and “the first”.

→ **Author comments:** This has been changed according to the suggestion.

15. lines 182 to 192: I would suggest to move this paragraph to SI.

→ **Author comments:** We have added the information about AgID to the figure caption of the sequence alignment in Fig. S3 (former Fig. S1).

16. Similarly, lines 193 to 194: please avoid “There are no structures of acceptor-product complexes available for any glycosyltransferases that use lipid substrates.” Instead, I would suggest to deeply discuss the impact of the PfdPMS-GDP-Man-Dol55-P complex in the understanding of other members of the GT2 family and/or other GTs.

→ **Author comments:** To accommodate this point, we have (as discussed in point 7) added topology diagrams for four GT2 members in Fig. S2 (two membrane GT2s and two soluble GT2s), and added comparisons with GtrB and ArnT in the Results section on p. 12 (“Structural comparison with related enzymes”).

17. lines 195 to 196: “however these structures are too different from true DPMSs to make a structural comparison meaningful.” I still suggest the authors to discuss more in deep and extensively the structural superposition/homology/differences of GtrB and PfDPMS in the context of enzymatic activity and fold:

- (i) r.m.s.d. value for equivalent residues of each protein
- (ii) location of the nucleotide sugar binding site in the GT-A domain
- (iii) donor substrate specificity
- (iv) importance of the juxtamembrane helices in both enzymes
- (v) catalytic mechanism

→ **Author comments:** At a superficial level *PfDPMS* and GtrB are indeed similar, but the fine details that distinguish *bona fide* DPMSs are lacking in GtrB, which we did not discuss in the original submission. A detailed comparison with GtrB has now been added in the Results section, including similarity scores (sequence identity, r.m.s.d. values etc.), as well as the topology diagram for GtrB in Fig. S2 and structural sequence alignment, subunit structure and active-site comparisons in Fig. S10.

18. Figure 4 should be divided as two different Figures: one referred to the catalytic mechanism, and a second one referred to the proposed model of substrate binding and product release – in the absence of experimental data supporting it – in the context of the “Discussion” section.

→ **Author comments:** The original Fig. 4 has been split into two figures: Fig. 4 (catalytic mechanism) and Fig. 6 (proposed model of substrate binding and product release). The discussion relating to substrate binding and product release changes have been moved to the Discussion section (see point 12 above).

19. Supplementary Table 1

- The following information should be introduced as first row in the Table: 5MLZ

(PfDPMS•GDP•Mg²⁺), 5MM0 (PfDPMS•GDP-Man•Mn²⁺), and 5MM1 (PfDPMS•GDP•Do155-P-Man).

→ **Author comments:** Supplementary Table 1 has been updated.

Reviewer #2 (Remarks to the Author):

Gandini and colleagues have determined structures of the dolichylphosphate β -D-mannose synthase from *P. furiosus*, in an empty unbound and with bound donor and acceptor substrates. In humans glycosylation is an important posttranslational modification, and aberrant glycosylation often has severe pathological consequences. The *P. furiosus* enzyme is predicted to be topologically similar to the human enzyme (on which no 3D-structural information is available), and, therefore, the results are also of great importance to understanding human glycosylation pathologies. The research has been done competently. The crystals used have high resolution (2.0 Å resolution for the empty enzyme), and thus the results appear solid. The description of the research, the results, and their implications is clear, as are the figures

Only one part is not very clear to me, and that is the experiment at 60 °C to trap the

product. What was the rationale? The methods are not very clear about whether only this experiment was done at 60 °C, or all crystallization experiments. Was crystallization at ambient temperatures not successful? Why was crystallization at 60 °C done, rather than doing it more slowly at room temperature?

Do the authors have data to support this approach? I am aware that catalysis is always better at higher temperatures because a larger fraction of the substrate molecules can attain the energy level of the transition state.

But crystallization is such lengthy experiment that I could imagine that temperature does not matter that much.

→ **Author comments:** When reading this comment, we realize that the temperature used for crystallization had not been sufficiently clarified in the Methods section. All crystallization experiments were performed at 4°C, not at 60°C. However, to obtain structure 5MM1 with the bound Dol-*P*-Man product in the active site, we used a reaction mixture that had been incubated at 60°C prior to crystallization.

The rationale for crystallizing a reaction mixture was to capture the reaction product, Dol-*P*-Man, and substrate turnover is somewhat higher at 60°C compared with lower temperatures. We screened many crystals made from reaction mixtures incubated at different temperatures, however we obtained only one that had sufficiently clear electron density for the glycolipid product.

Since crystallizing a wild-type enzyme that has been allowed to react with its natural substrates will inevitably produce a number of enzyme (conformational) states (i.e. the enzyme caught at different steps of the process) as well as a mixture of more or less completely turned over substrate/products, this type of strategy leads to protein crystals whose content is “convoluted” - the electron density contains traces of the different states and we may only distinguish with some certainty the most occupied states with respect to conformations and products.

Furthermore, the product will eventually be released, which further hampers the possibility to capture it bound to the enzyme. This explains the general low success rate of finding crystals with authentically bound product, which is why many crystals had to be screened and different temperatures tested. Our interpretation is that at higher temperatures, substrate conversion is likely to be more complete. Possibly, a similar result can be achieved with lower temperature but increasing the incubation time. Working with dolichol-based substrates is demanding for several reasons (expensive, laborious to prepare etc), and although alternative strategies may also work, it was not practically feasible to do an exhaustive screening of conditions.

Additional changes:

Besides other (minor) changes/corrections/optimizations not requested by the referees, we have also :

- Added a section in the Discussion that relates to polyisoprenol-recognition sequences (PIRS). To accompany this discussion, a new figure (Fig. S12) has been added to the supplement, which highlights the regions in *P*/DPMS that interact with the dolichylmonophosphate portion of the Dol-*P*-Man, and points out which of these residues are conserved in the human enzyme.

- As part of the comparison with GtrB in the Results section, we also performed an additional experiment where different types of donors were tested with *PfDPMS* in combination with Dol55-*P* as acceptor (i.e. GDP-Man, GDP-Glc, UDP-Glc). GtrB use UDP-Glc, and we wanted to assess the level of promiscuity of *PfDPMS* for alternative donors. The results are presented on p. 14 and in Fig. S11, and show that *PfDPMS* is specific for GDP-Man, with some residual activity with GDP-Glc, but virtually incompetent with UDP-Glc. The authors of the GtrB article did not report results for donors other than UDP-Glc, but our results confirm the general understanding of GTs as relatively promiscuous enzymes.
- Additionally, we removed former Fig. S5.

Reviewers' Comments:

Reviewer #1 (Remarks to the Author):

Thank you very much for taking into account all my comments, which have been fully addressed. Therefore I support the publication of this manuscript in Nature Communications. I would like to take this opportunity to congratulate the authors for this very nice and elegant contribution.

Reviewer #2 (Remarks to the Author):

The authors have addressed my comments to my complete satisfaction. The manuscript has much improved, and I found it a joy to read.